# Recent Progress in the Discovery and Development of Monoclonal Antibodies against Viral Infections

**DOI:** 10.3390/biomedicines10081861

**Published:** 2022-08-02

**Authors:** Pardis Mokhtary, Zeinab Pourhashem, Akram Abouei Mehrizi, Claudia Sala, Rino Rappuoli

**Affiliations:** 1Monoclonal Antibody Discovery Laboratory, Fondazione Toscana Life Sciences, 53100 Siena, Italy; p.mokhtary@toscanalifesciences.org; 2Department of Biochemistry and Molecular Biology, University of Siena, 53100 Siena, Italy; 3Student Research Committee, Pasteur Institute of Iran, Tehran 1316943551, Iran; z.pourhashem@yahoo.com; 4Malaria and Vector Research Group, Biotechnology Research Center, Pasteur Institute of Iran, Tehran 1316943551, Iran; abouei@gmail.com

**Keywords:** monoclonal antibody, mAb, viral infections

## Abstract

Monoclonal antibodies (mAbs), the new revolutionary class of medications, are fast becoming tools against various diseases thanks to a unique structure and function that allow them to bind highly specific targets or receptors. These specialized proteins can be produced in large quantities via the hybridoma technique introduced in 1975 or by means of modern technologies. Additional methods have been developed to generate mAbs with new biological properties such as humanized, chimeric, or murine. The inclusion of mAbs in therapeutic regimens is a major medical advance and will hopefully lead to significant improvements in infectious disease management. Since the first therapeutic mAb, muromonab-CD3, was approved by the U.S. Food and Drug Administration (FDA) in 1986, the list of approved mAbs and their clinical indications and applications have been proliferating. New technologies have been developed to modify the structure of mAbs, thereby increasing efficacy and improving delivery routes. Gene delivery technologies, such as non-viral synthetic plasmid DNA and messenger RNA vectors (DMabs or mRNA-encoded mAbs), built to express tailored mAb genes, might help overcome some of the challenges of mAb therapy, including production restrictions, cold-chain storage, transportation requirements, and expensive manufacturing and distribution processes. This paper reviews some of the recent developments in mAb discovery against viral infections and illustrates how mAbs can help to combat viral diseases and outbreaks.

## 1. Introduction

Viruses are microorganisms characterized by a wide range of features, such as shape, size and infectivity, and can cause mild to severe human and animal disease. Appendix A lists rare to pandemic viruses.

Our immune system has evolved strategies to neutralize viruses in various ways. The first contact between the virus and the cell host membrane represents the initial challenging step in the viral infectious life cycle. Antibodies, produced in response to virus detection, may act as a barrier, thereby preventing the virus from completing this crucial step. Antibodies may also clear viruses from the body before they have the possibility to enter a cell. They neutralize the pathogen by binding to free viruses (opsonization) and therefore blocking the interaction between the virus and the host cell. Exposure of portions of viral proteins (i.e., epitopes) on the cell surface through the major histocompatibility complex I (MHC I) allows T cells to kill infected cells. Other killing mechanisms are mediated by antibodies. These are antibody-dependent cellular cytotoxicity (ADCC), antibody-dependent cellular phagocytosis (ADCP), and complement-dependent cytotoxicity (CDC). These mechanisms help contain and clear the viral infection [1] (Figure 1). ADCC is the nonphagocytic killing of an antibody-coated target cell by a cytotoxic effector cell. The mechanism involves the release of cytotoxic granule content or the production of cell death-inducing molecules. The interaction of target-bound antibodies (IgG, IgA, or IgE classes) with specific Fc receptors (FcRs), glycoproteins on the effector cell surface that bind the Fc portion of immunoglobulins(Ig), triggers ADCC [2]. Monoclonal antibodies can enhance ADCC activity through their Fc portion, whose glycosylation pattern was shown to impact this effector function [3]. ADCP is a kind of cell-mediated immunity in which immune system cells phagocytose target cells or pathogens that have been bound by specific antibodies. In this case, the antibody Fc region engages with Fc receptors exposed on the surface of phagocytic cells and causes engulfment and killing [4]. Finally, CDC is a robust effector mechanism that consists in antibody binding to the complement component C1q to initiate the classical complement cascade, resulting in the assembly of the membrane attack complex (the complement cascade’s cytolytic end product) and lysis of the antibody-targeted infected cells [1].

Antibodies are increasingly considered as an innovative and valuable class of therapeutic agents because of their unique target specificity, which promotes infection clearance [5]. Monoclonal antibodies (mAbs) derive their name from the clone of white blood cells which produced them and have gained interest in recent years as they can find application in diverse areas including medicine and biotechnology. Importantly, mAbs have been proposed as medications against viruses such as HIV and influenza and have recently been exploited in COVID-19 prophylaxis and therapy [5,6].

The mAb field has seen considerable progress since Emil von Behring and Shibasaburo Kitasato discovered antibodies in the 1890s. mAbs have the ability to target a wide variety of microbial organisms, including bacteria, viruses, parasites, fungi, and toxins [7]. Various mAbs are being used to conduct clinical trials against pathogenic viruses, including HIV and SARS-CoV-2 which, together, account for more than 40% of clinical trials (Figure 2). Other viral infections may be tackled with mAbs in the near future. For instance, influenza (10% of total clinical trials), Epstein−Barr virus (8% of total clinical trials), hepatitis C (8% of total clinical trials) and respiratory syncytial virus (7% of total clinical trials) (Figure 2). In this article, we will review the most recent advances, challenges and opportunities in mAb discovery and development against viruses.

## 2. Progress in Developing Antiviral mAbs

In 1971 the virologist and Nobel laureate David Baltimore proposed a virus classification system based on the way viruses synthesize their messenger RNA (mRNA). The so-called Baltimore classification separates viruses into seven groups according to their nucleic acid content (DNA or RNA), whether the genome is single- or double-stranded, and the sense (positive or negative) of the RNA genomes [8]. The Baltimore classification integrates virus taxonomy, which relies on evolutionary history instead. In this section we will present examples of viruses belonging to various groups and of the most recent advances in the discovery and development of mAb-based therapies.

**Zika virus** is a member of the *Flaviviridae* family with positive-sense single-stranded RNA genome (Baltimore Group IV) and is related to yellow fever, dengue, and West Nile viruses. Zika is a mosquito-borne pathogen that has become a major public health problem, posing a threat to more than two billion people. The disease caused by Zika is especially relevant in pregnant women, where it can cause severe brain malformation in the fetus [9,10].

Integration of different approaches is required for reducing the spread of the virus and the risk of infection. For instance, research into the ecological niche of the female *Aedes aegypti* mosquito has contributed to the identification of the areas at risk [11,12]. On the therapeutic side, small molecule inhibitors and antibodies have shown promise in mouse models [13]. For instance, Rianne N. Esquivel and colleagues proposed synthetic DNA-encoded monoclonal antibody (DMAb), which allows very powerful mAbs to be delivered in vivo to control Zika virus infections. The neutralizing antibody DMAb-ZK190 has been designed to bind to a protein on the Zika virus called ZIKVE. In vivo findings demonstrated that this synthetic antibody rapidly protects mice and monkeys from Zika and may control infection. The candidate mAb is currently in clinical trials [13].

Another study worked on recombinant antibodies against the Zika virus, for at-risk individuals, to provide a feasible alternative to vaccinations. Recombinant mAbs might well be designed to avoid infection enhancement; they may be a safer and effective alternative to vaccinations for providing fast protection. A study showed that administration of two antibodies, Z004 and Z021, to pregnant macaques protects the fetus from neurologic impairment and reduced Zika virus vertical transmission [14]. Another work reported that a cocktail of three different neutralizing human mAbs, which target distinct epitopes of the pathogen and have been engineered to abolish antibody-dependent enhancement of disease (ADE), can prevent Zika virus infection in challenged macaques [15].

**Ebolavirus** (EBOV) is a member of the *Filoviridae* family with negative-sense single-stranded RNA genome (Baltimore Group V) that causes Ebola virus disease (EVD). The frequent outbreaks registered mostly in Central and West Africa prompted researchers to focus their efforts on the discovery of mAbs and of small molecules that can be effective and easily produced in emergency situations. There are at least five Ebolaviruses known so far; most of them infect animals and humans, causing hemorrhagic fevers with high fatality rates [16]. Zaire EBOV is the most dangerous of the known EVD-causing viruses and is responsible for the largest number of outbreaks [17,18]. Several antibodies have demonstrated good protection in animal models, but few of them are in clinical trials [19].

The EBOV surface glycoprotein (GP) is the primary target of antibody-based treatment and vaccination because it is expressed on the viral surface and enables virus attachment to and entry into host cells. Furin cleaves GP into disulfide-linked GP1 and GP2 subunits, which then assemble into metastable trimers once within the cell surface. The GP1 subunit is responsible for cellular attachment and binding to the Niemann Pick C1 (NPC1) receptor, while the GP2 subunit is necessary for viral and cellular membrane fusion [16,20].

Several GP-targeting mAbs and mAb cocktails have been discovered and produced in recent years. Among them, ZMapp [21], MIL77E [22], mAb114 [23], and REGN-EB3 [24] proved to be highly effective [19].

mAb114 is a single mAb that targets the receptor-binding domain of the EBOV GP and prevents mortality in rhesus macaques after a lethal challenge with Zaire EBOV. It is the only mAb obtained from a human survivor of EVD tested as a therapeutic drug [25]. After mAb114, other mAb-based therapeutic options followed, including REGN3470-3471-3479 and ZMapp [25]. Ridgeback Biotherapeutics produced Ansuvimab (ansuvimab-zykl; EBANGA), a human mAb that binds to Zaire EBOV GP to prevent it from entering host cells. Ansuvimab acquired initial clearance in the United States on 21 December 2020 [26].

In vitro, Ebola-specific antibodies promote antibody-dependent cellular cytotoxicity (ADCC) in human peripheral blood natural killer (NK) cells and NK cell lines; thus, Fc-mediated involvement in anti-Ebola mAb therapy and vaccine-induced safety is gaining increasing interest (5).

The possibility to combine mAbs into a cocktail represents a promising avenue that deserves further exploration and exploitation. However, in order to rationally design successful mAb cocktails, the precise epitope recognized by different mAbs must be identified by means of genetics and/or structural biology. In addition, novel epitopes and key residues of GP should be considered, such as Q206 and Q411, which were shown to be targeted by new potential candidate mAbs for EBOV prevention and treatment [19].

MB-003 (human or human-mouse chimeric mAbs c13C6, h13F6, and c6D82) is amongst the most potent and effective combinations [27]. Another cocktail is ZMapp, composed of m1H3, m2G4, and m4G7 murine mAbs, expedited for clinical trials during the 2013–2016 EBOV pandemic in West Africa, with encouraging preclinical evidence [28,29]. Interestingly, ZMapp—a mixture of three mAbs that are not protective if administered individually—was shown to protect non-human primates from a fatal EBOV challenge [29,30].

Researchers created plasmid DNA-encoded mAb (DMAb) that encode powerful anti-Zaire EBOV glycoprotein (GP) mAbs from EVD survivors. They discovered that DMAb-11 and DMAb-34 have functional and molecular characteristics similar to their recombinant counterparts, have a broad expression window, and provide fast protection against deadly EBOV exposure in mice [31].

**Influenza virus** is a member of the *Orthomyxoviridae* family with a negative-sense single-stranded RNA genome (Baltimore Group V). According to the World Health Organization, annual epidemics result in 2–5 million severe cases and 250,000 to 500,000 fatalities worldwide [32].

Vaccination is one of the most effective ways to prevent influenza. The trivalent inactivated influenza vaccine, live attenuated influenza vaccine and subunit influenza vaccines are used in the clinic [33,34]. Nevertheless, the efficacy of the influenza vaccines may be significantly attenuated and can be invalid if a large mismatch exists between the vaccine and the epidemic strain. Broadly neutralizing antibodies (bnAbs) against influenza viruses would thus represent an ideal option in influenza prevention and therapy [35,36,37]. Choosing the most appropriate strain(s) for vaccine manufacturing is challenging because of antigenic drift and of the wide variety of possible emerging zoonotic and pandemic viruses. Therapeutic approaches or passive immunization [38] based on mAbs targeting the conserved exposed epitopes on the virion suffer from the same issues.

The most frequently targeted antigen is perhaps hemagglutinin (HA). Other viral antigens are NP and M1. Antibodies against the latter are usually more difficult to obtain and are non-neutralizing, as M1 is an internal protein that is seldom exposed outside of viral particles. As a result, the emphasis has been on inducing cytotoxic T-cell responses [38,39].

A highly specific anti-influenza A antibody therapy with MHAA4549A is being developed to meet the essential clinical need of treating hospitalized patients with severe influenza A. MHAA4549A is a human immunoglobulin G1 (IgG1) mAb that binds to a highly conserved epitope on the stalk of influenza A HA and neutralizes all known human influenza A strains [40]. Efficacy and safety of this antibody have been addressed in Phase 1 and Phase 2 clinical trials. MHAA4549A is safe and well-tolerated in healthy individuals up to a single intravenous dose of 10,800 mg. It has linear serum pharmacokinetics, typical of a human IgG1 antibody with no known endogenous targets. In a Phase 1 trial and Phase 2a research, the anticipated clearance and projected effective dosages for MHAA4549A were confirmed in people [41,42].

bnAbs, which target most of the circulating influenza viruses with pandemic potential, might be a viable short-term preventative or therapeutic alternative. One of the first isolated mAbs that can broadly neutralize is C179. It can neutralize the H5, H6, and H9 strains [43]. Mice infected with H1N1, H5N1, H3N2, and influenza B viruses were shown to be protected by CR6261 and CR9114 [44]. CR9114 is a bnAb capable of inhibiting influenza A and B viruses. Even though CR9114 did not have in vitro neutralizing activity against the human H2 virus, Sutton and colleagues showed that both CR6261 and CR9114, mAbs that bind the stem region of the HA molecule, are effective against infection with H2 viruses of both human and animal origin in mice. These observations highlight the significance of in vivo evaluation when testing bnAbs [44,45]. Compared to other broadly neutralizing mAbs that cross-neutralize several different subtypes of influenza A virus, such as CR6261, F10, 12D1, CR8020, and FI615, CR9114 is one of the most broadly neutralizing antibodies that have been identified to date [43,45].

The extracellular domain of Matrix protein 2 (M2e) is another highly conserved interesting target for mAbs against influenza viruses, although it is considered as a subdominant epitope [46]. TCN-032, an entirely human mAb that targets the ecto-domain of M2, was isolated and shown to be cross-reactive [47].

Engineering the chimpanzee adenovirus AdC68 to express CR9114 resulted in AdC68-CR9114, whose passive protective activity was evaluated in vitro and in vivo [43]. AdC68-CR9114-infected cells expressed the bnAb at high level in vitro and in vivo, exhibited biological functions, and protected mice from different types of influenza virus [43].

Clinical studies including bnAbs were encouraging. For instance, VIS410, an anti-HA mAb that binds to the stem of the protein, was shown to be safe, well-tolerated, and efficacious in reducing nasopharyngeal viral load and viral shedding [48]. Moreover, anti-M2e antibodies (TCN-032) were widely employed to neutralize the influenza virus, and the virus titer in patients was considerably decreased [49,50].

**SARS-CoV-2 virus** is a member of *the Coronaviridae* family with a positive-sense single-stranded RNA genome (Baltimore Group IV), causing the ongoing pandemic severe acute respiratory syndrome named coronavirus disease 2019 (COVID-19). The enveloped RNA virus enters host cells by binding to the angiotensin-converting enzyme 2 (ACE2) receptor through the Spike (S) protein [51].

Even though vaccines are the best strategy for COVID-19 prevention, mAbs can be used as therapeutic agents, especially in vulnerable populations such as the elderly, the immune-depressed and those who cannot be vaccinated for various medical reasons [52]. Fifty-nine antibodies, including 20 neutralizing antibodies, and 178 vaccines were under development at different stages in a survey completed on 21 February 2022, according to the article “Biopharma products in development for COVID-19” released by *Bioworld* [53,54].

Antibodies can inhibit viral entry by blocking the viral S protein or host cell receptors/co-receptors on the respiratory system, gastrointestinal tract, and endothelium cells, thus inhibiting virus attachment to the targeted host cell. Secondly, antibodies act as immune-mediators to control damage caused by hyper-activation of the host immune response, which results in exacerbated inflammation and poses a severe risk of death [55].

To date, several human, humanized, or bioengineered mAbs for therapeutic use targeting different parts of S-protein of SARS-CoV-2 have been authorized for emergency use (EUA) based on Phase 1/2 and Phase 2 data. Sequence identity between SARS-CoV (which caused the SARS epidemic in 2002–2004) and SARS-CoV-2 S proteins allowed the repurposing of several antibodies which had been previously found to interact with SARS-CoV by binding to the receptor-binding domain (RBD), such as CR3002, F26G19, 2B2, 1A9, 4B12, 1G10, and S309. Interestingly, S309 neutralizes SARS-CoV-2 more potently than SARS-CoV [56].

REGN-COV2 is one of the first mAb cocktails approved for use against SARS-CoV-2 infection. It consists of two potent neutralizing IgG1 mAbs with unmodified Fc regions, namely casirivimab and imdevimab, which bind to non-overlapping epitopes of the RBD. This combination therapy by Regeneron Pharmaceuticals was approved for EUA by the FDA on 21 November 2020, at a dose of 2.4 g (1.2 g casirivimab and 1.2 g imdevimab), intravenous (IV) administration for treatment of high-risk patients [57]. Other candidates, including MAD0004J08, isolated from a convalescent COVID-19 patient, were shown to bind and interfere with the RBD/ACE2 interaction with all tested viral variants and are currently being evaluated in clinical trials [58,59].

Bispecific antibodies strategically combine two different antibody specificities into one single molecule. They have been proposed as new tools to treat COVID-19 [60]. bsAb15 is a bispecific monoclonal antibody (bsAb) based on B38 and H4 and has greater neutralizing efficiency than the two single parental antibodies [61]. CoV-X2 was developed starting from C121 and C135, derived from donors who had recovered from COVID-19, and showed greater efficacy in neutralizing wild-type SARS-CoV-2 and its variants of concern and escape mutants generated by the parental mAbs [62].

The Eli Lilly combination of LY3819253 (LY-CoV555) with LY3832479 (LY-CoV016) and VIR-7831/GSK4182136 (Sotrovimab) are two anti-spike candidate medications in Phase 3 clinical trials. Another combination therapy is the AbCellera Biologics and Eli Lilly Bamlanivimab, composed of bamlanivimab and etesevimab, authorized for EUA and administered only to newly diagnosed mild/moderate COVID-19 patients. This mAb cocktail significantly decreased viral load and fewer COVID-19-related hospitalizations and deaths were observed in the treatment group compared to the control cohort [63]. For a complete list, updated in March 2022, please refer to Table 1.

Further evolution and spread of viral variants such as the variants of interest and concern of SARS-CoV-2, the current Omicron (B.1.1.529) and Omicron-2 (BA.2), raised serious concerns about resistance to vaccine-induced neutralizing antibodies and have challenged diagnostics, treatment, and vaccine development. The emergence of variants in SARS-CoV-2 will require new formulation of vaccines and therapeutics to consider the numerous mutations in the S protein, which could cause escape from the antibody-mediated neutralization and increase the risk of reinfections. Based on a CDC December 2021–June 2022 report [64], fortunately, vaccines approved and authorized for use in the United States are still effective against the predominant variant circulating in the United States, and efficacious therapeutics are available [65,66,67].

Special mention should be dedicated to nanobodies, i.e., single-chain antibodies consisting of a single variable domain. Nanobodies are much smaller than human mAbs and are usually found in camelids like alpacas and llamas. They have been proposed as anti-tumor therapeutics [68] and have found recent application in anti-SARS-CoV-2 research projects [69]. Nanobodies have certain advantages compared to mAbs. They can be easily produced in prokaryotic and eukaryotic expression systems, may have superior penetration properties, can be conjugated to drugs, can be exploited as molecular probes, and potentially can represent a way to efficiently manufacture biotherapeutics on a large scale [70].

The second group of mAbs developed for COVID-19 therapy includes anti-cytokine mAbs, which target the cytokine and chemokine storm, including IL-2, IL-6, IL-7, IL-17, G-CSF, GM-CSF, IP-10, MCP1, MIP1A, TNFα, IFN-ɣ, VEGF, and CCL2, described in patients with severe COVID-19 [71].

Clinical trials in different phases of anti-cytokine/chemokine mAbs are currently ongoing in COVID-19 patients, including of Tocilizumab (IL-6 inhibitor), Lenzilumab (a GM-CSF antagonist), Risankizumab (humanized mAb against interleukin-23), Gimsilumab (KIN-190 N), Mavrilimumab/KPL 30D, and TJ003234 (three other anti-GM- CSF mAbs), Leronlimab (CCR5 antagonist), Canakinumab (Anti IL-1β), CPT-006 and AK119 (Anti CD73), Garadacimab/CSL312 (Factor XIIa antagonist), Pamrevlumab (mAb against connective tissue growth factor), Bevacizumab (Anti VEGF), Cizanlizumab (Anti P-selectin), Ravulizumab (Anti C5), and Emapalumab (IFNɣ antagonist) and Anakinra (IL-1 antagonist). A complete list with a brief description of candidate mAbs can be found at COVID-19 Biologics Tracker [72].

**The Middle East respiratory syndrome coronavirus** (MERS-CoV) is a member of the *Beta coronavirus* genus with a positive-sense single-stranded RNA genome (Baltimore Group IV). MERS-CoV was first reported in June 2012 and is considered as one of the viruses identified by WHO as a likely cause of a future epidemic (http://www.who.int/emergencies/mers-cov/en/, accessed on 30 June 2022).

The functional and structural significance of the MERS-CoV S protein makes it an important antigenic candidate for neutralizing-antibody-mediated protection against CoVs. The S protein includes the N-terminal domain (NTD), receptor-binding domain (RBD), HR1 and HR2 in the S2 subunit. Most mAbs against both SARS-CoV and MERS-CoV target the RBD in the S protein to prevent virus attachment to the host cell [73].

There is a predominance of RBD-interacting mAbs for MERS-CoV, such as LCA60 (which recognizes neutralizing epitopes of S1-NTD) [74], MERS-4, MERS-27 (which recognizes neutralizing epitopes of the RBD) [75,76], m336, m337, and m338 (which recognize neutralizing epitopes of RBD overlapping with the DPP4-binding site) [77,78,79]. Humanized neutralizing antibodies derived from mice have been reported as well. For instance, 4C2 and 2E6 showed promising activity against the virus both in vitro and in vivo [80].

80R, m396, CR3014, and S230.15, produced against different strains of SARS-CoV, target epitopes in the RBD of the S protein of SARS-CoV as well as of MERS-CoV [81]. REGN3048 and REGN3051 block S-mediated pseudotyped MERS-CoV entry, neutralize infection of divergent strains of pseudotyped and live MERS-CoV and are undergoing a Phase 1 clinical trial [82].

**The human immunodeficiency virus (HIV)** is a member of the *Retroviridae* family with a positive-sense single-stranded RNA genome (Baltimore Group VI). It causes acquired immuno-deficiency syndrome (AIDS), a condition in which progressive failure of the immune system allows life-threatening opportunistic infections and cancers to thrive [83,84].

Research has shown that reducing the level of the virus in the blood and increasing immunity to the virus could be achieved by using neutralizing mAbs. In addition, mAbs can play a relevant role in inducing apoptosis and death of infected cells in laboratory models [85,86,87]. The new generation of mAbs against the HIV-1 virus envelope protein Env is highly effective in preventing multiplication of the virus [88,89]. HIV-1 specific mAbs targeting the Env protein are listed in Appendix A.

A trial designed to test the preventive effects of VRC01, a broadly active mAb directed against the CD4 binding site of HIV-1, is ongoing in high-risk populations [90]. An antiretroviral therapy (ART) rollout has led to a massive reduction in AIDS-related deaths [91]. The benefit of adding bnAbs to ART has gained interest in recent years. However, no single bnAb can neutralize all HIV-1 circulating viral variants [92,93]. Novel isolated and bioengineered antibodies with greater potency, breadth, and longer half-life than the prototypic antibodies (e.g., VRC01 and 3BNC117) will probably increase immunotherapy efficacy and reduce treatment costs. The challenge for the research field now is to optimize all available resources and design more effective treatment regimens while vaccine development efforts continue [94,95,96]. A Phase 1 study with infants born to HIV-infected mothers will be initiated shortly. If the efficacy of these three antibodies (VRC01, VRC01LS, VRC07-523LS targeting CD4 binding site) against HIV-1 transmission is shown in this study, it would be the first in vivo proof that human mAbs are a significant determinant in the development of a therapeutic vaccine. Research further includes the promise of bispecific and trispecific bnAbs [90,97]. Glycosylphosphatidylinositol (GPI)-anchored single-chain fragment variable (ScFv) might be utilized as an overall and successful method for identifying antibodies reacting to HIV-1 envelope proteins and other enclosed viruses with temporarily exposed neutralization epitopes [98]. GPI-anchored proteins are transported to the plasma membrane’s lipid rafts. The lipid raft has been considered an entry point for HIV-1 [99,100,101,102]. Utilizing a GPI anchor in the mRNA construct of mAbs and expression on the surface may increase the efficiency of antibodies by preventing neutralizing antibodies from dispersing from virus-infected tissues. Moreover, the GPI anchor may boost the local antibody concentration in the target organ and, as a result, enhance antibody persistence on the mucosal surface [103,104]. Recent studies showed that mRNA-produced anchored neutralizing antibodies, both full and single domain, are quickly expressed and persist on the surface of transfected cells in culture and the lungs of mice. This technology has also been utilized to treat HIV-1 infections [105].

Similarly to what happens with antibiotic use, treatment of infectious diseases with mAbs may result in the selection of resistant mutants which are no longer susceptible to the medication. For example, it was shown that a mutant version of HIV could be isolated using only one mAb in an in vivo experiment [106,107]. A way to deal with the risk of selecting resistant mutants consists in combining several mAbs in a cocktail therapy [108]. For instance, the simultaneous use of 4E10, 2F5, and 2G12 may prevent mother-to-child transmission of HIV and be effective as post-exposure prophylaxis after accidental exposure to HIV-1-contaminated materials [108,109].

Glycoengineering also has a good prospect for HIV therapy. HIV-specific bnAbs produced by glycoengineered-adeno associated virus (GE-AAV) vectors were analyzed for fucose content and ADCC [110]. Glycoengineered AAV-transduced cells generated the antibody, which was subsequently modified to remove α1–6 fucose. Engineered antibodies boosted HIV-infected target cells’ ADCC by 40–60% [110].

**Human cytomegalovirus (HCMV)** is a member of *Herpesviridae* with a double-stranded DNA (dsDNA) genome and classified as a Group I member in the Baltimore classification system [111]. HCMV is another virus with no approved vaccine yet. Despite this, vaccine research is progressing well, with many candidate vaccines now being evaluated [112]. A combination of antivirals and passive vaccination with HCMV hyperimmune globulin (CMV-HIG) demonstrated some benefits [113]. HCMV mAbs exhibit superior neutralizing efficacy as compared to polyclonal CMV-HIG antibodies, according to a recent in vitro investigation [114,115]. However, the only FDA-approved antibody prophylaxis for HCMV infection is CMV-HIG [116]. Antibodies suppress HCMV in a different way than small molecule antivirals such as ganciclovir, cidofovir, foscarnet, and letermovir; therefore, they should be a good complement in the clinic, either alone or in conjunction with antivirals [117]. Anti-HCMV antibodies used in conjunction with antivirals may help overcome medication resistance and lessen adverse effects by decreasing the antiviral drug dose [118]. Novel neutralizing antibodies that correlate well with in vivo protection against HCMV are one method to improve existing antibody combinations. Neutralizing antibodies that target the viral glycoproteins gH/gL/gB represent promising molecules [119]. HCMV gB is a type III viral fusion protein that interacts with the gH/gL heterodimer to form the viral envelope [120]. A high titer of gB AD-2 specific antibodies has been linked to protection against HCMV infection or illness in several investigations [121,122,123].

Both the gB AD-4 specific LJP538 and the pentamer-specific LJP539 in CSJ148 (Novartis) are substantially more powerful in neutralization than CMV-HIG (CytoGam1) [124]. Antibodies that target the gH protein neutralize HCMV. MSL-109 is a gH-specific human mAb discovered in the late 1990s which prevents gH/gL dimerization. MSL-109 was well tolerated and safe in allogeneic hematopoietic stem cell transplantation (HSCT) patients; however, it did not lower HCMV infection or illnesses [125]. The Phase 2/3 trial was suspended [125,126]. RG7667 (Genentech) comprises a humanized mouse antibody (MCMV3068A) that targets the pentamer (a multi-protein structure required for viral attachment to cells [127]) and an affinity-matured form of MSL-109 (MCMV5322A) [127]. The MCMV5322A mAb targets gH, while the MCMV3068A mAb targets the pentamer complex. Despite the positive results, RG7667 research was halted in a Phase 2 trial [128]. 13H11 is a mAb that binds to gH in a different way than MSL-109, thus making it attractive for use as a cocktail component [129]. TCN-202 is a gB AD-2 site I-specific human mAb produced by Theraclone Sciences. It was considered as a promising candidate in Phase 1 research but was never further developed [130]. Other mAbs are still in the preclinical stage, including: anti-CD3/anti-gB BiTE [131,132], r272.7, and r210.4 [133]. They are anti-gB, whilst anti-CD3/anti-gB BiTE is designed to redirect non-specific T cells for the focused destruction of HCMV infected cells [133]. On the other hand, complement enhances the in vitro neutralization activity of r272.7 and r210.4 [133]. Two additional promising antibodies are BsAb-F1 and BsAb-F2. Both of them are IgG-ScFv-based bispecific-neutralizing antibodies with broad cell type coverage [134]. 

**Respiratory syncytial virus (RSV)** is a member of the *Pneumoviridae* family with single-stranded, negative-sense RNA discovered in 1955 [135,136]. Various promising mAbs and RSV vaccines are now being tested in clinical trials, with the goal of providing protection to the most susceptible groups [137]. Antibody-based medicines against RSV include IGIV (Respigam) [138], palivizumab (Synagis) [139], and MEDI-524 (Numab) [140]. Of note, Synagis was the first mAb successfully developed to combat an infectious disease. Unfortunately, their use will be restricted due to their high cost [141].

The fusion (F) and attachment (G) glycoproteins are critical for infectivity and viral pathogenesis because they contain the antigenic determinants that evoke neutralizing antibodies [142]. In the late 1990s and early 2000s, RespiGam, an RSV intravenous immune globulin infusion, was used prophylactically to avoid severe RSV-associated lower respiratory tract illness in young children with bronchopulmonary dysplasia or preterm delivery. In 2003, the usage of RespiGam was phased out in favor of palivizumab prophylaxis (Synagis) [143].

Synagis (palivizumab) is a mAb from a humanized mouse that binds to the F glycoprotein. It has been authorized by the FDA [144]. The use of a humanized mAb against the RSV fusion (F) protein for immunoprophylaxis has been shown to reduce the risk of hospitalization due to severe RSV illness [134,144]. However, escape mutations might change the vulnerability of wild-type RSV strains missing the mAb epitope. This possibility highlights the need of conducting surveillance studies before, during, and after mAb clinical trials to determine the impact of the new mutations on the immune protection provided by the mAb [137].

Antibodies that are 10–50 times more powerful than palivizumab have been discovered, and many of them identify epitopes present solely on the prefusion conformation [145].

Nirsevimab (MEDI-8897) is a human neutralizing IgG1K antibody that targets the pre-F antigenic region. Three amino acid changes in the Fc region of the IgG heavy chain enhance its half-life (YTE technology) [137].

Research on MEDI-8897 mAb suggests that a single dose may be enough to protect neonates from severe illness throughout the RSV season, and results from a Phase 2b clinical study were due in late 2018 [146]. D25, a human antibody, was improved to create MEDI-8897, a variation with increased potency and a longer serum half-life [147]. Other prefusion-specific antibodies, as well as antibodies directed against the G protein that impede viral attachment, are in the early stages of research [148,149]. 

There are mAbs against RSV F and G antigens, and cocktails of antibodies against these two components are also being studied, which might lead to synergistic effects and decreased viral escape [135]. One strategy for improving results is to extend the life of antibodies. For example, MK-1654 (Merck & Co; Inc.) is a long-acting mAb that is now in Phase 1 clinical trials [137,150]. 

Suptavumab, an RSV antibody that has shown promising results in vitro and in animal models, is 40 times significantly more efficient than palivizumab in neutralizing RSV. Unfortunately, the study’s clinical progress was halted [146,150]. Several antiviral medicines, vaccines, and mAbs with extended half-life are now in clinical testing; nevertheless, market availability is anticipated to take several years [151]. Hopefully the present excitement around RSV therapies will allow one or more successful interventions to be approved in the coming decade [152].

## 3. Standard and New Technologies for the Discovery and Development of Highly Effective mAbs

The recent technical progress in B cell sorting, sequencing, and cloning accelerated the mAb discovery and development process. In this section we will review the most recently developed methodologies which contribute to feeding the mAb discovery pipeline with increasingly safe and powerful candidates. 

The **hybridoma method** is the most widely used technology for producing therapeutic antibodies from nonhuman sources. It consists in isolating B lymphocytes from mice that have been immunized with an antigen of interest and in fusing them with immortal myeloma cells to form hybrid cells, i.e., the hybridoma cells, which can then express mAbs against the specific antigen [153]. The advantage of this well-validated technology consists in the possibility to generate mAbs against virtually any antigen of interest. On the other hand, fusion efficiency can be low and mAbs obtained by the hybridoma method are usually nonhuman and this may be associated with downstream issues in the effector functions. Some examples of mAbs produced by means of the hybridoma methodology are presented below.

Eculizumab, a high-affinity humanized mAb that binds to the complement protein C5, prevents formation of the terminal complement complex C5b-9, which is involved in cell lysis, by inhibiting cleavage of C5 to C5a and C5b. By retaining early complement components, the C5 blockade has an indirect immunoprotective and immunoregulatory effect [154]. Leronlimab is a humanized IgG4 mAb directed against the C-C chemokine receptor type 5 (CCR5) which is being investigated as a therapy against HIV and various types of cancer. Other examples are represented by West Nile virus MGAWN1 (a neutralizing humanized mAb to West Nile virus E protein [155], MEDI-524 (Motavizumab), a humanized mAb with enhanced potency against respiratory syncytial virus (RSV) [140,156], KD-247, an anti-V3 humanized antibody that suppresses human immunodeficiency virus type 1 ex vivo and provides monkeys with sterile defense against a heterologous simian/human immunodeficiency virus infection [157]. KD-247 has the potential to be useful not only as a passive immunization antibody for HIV prevention but also as immunotherapy for HIV suppression in phenotype-matched HIV-infected people [157].

In addition to the hybridoma method, other **B cell immortalization methods** exist and have been used for mAb production. The Epstein−Barr virus (EBV) can transform and immortalize B cells, thus ensuring rapid screening of candidate mAbs [158,159]. Other immortalization tools include expression of BCL-6 and BCL-XL (anti-apoptotic Bcl-2 protein family). Introduction of these genes into peripheral blood memory B cells generates highly proliferating cells which secrete mAbs [160]. BCL-6 and BCL-XL transduced cells express the enzyme activation-induced cytidine deaminase (AID), which mediates somatic hypermutation and class-switch recombination and therefore increases diversity of the mAb repertoire. Immortalized cells can be maintained in culture for a long time.

**The phage display** technique relies on using bacteriophages (i.e., viruses that infect bacteria) to express a unique protein variant (such as antibody fragments). A gene encoding the protein of interest is implanted into a phage, allowing the phage to display the protein on the surface. The so-called phagemid plasmid, a recombinant phage display plasmid, improves the possibility of expression of both chains of target antibodies. Phage display benefits from the relatively easy manipulation of phagemid plasmids and this partially compensates for limitations imposed by phage size and proteins that can be accommodated on its surface. This system has been used to isolate antibodies which can neutralize a spectrum of viruses such as severe acute respiratory syndrome coronavirus (SARS-CoV), Ebola virus, yellow fever virus, hepatitis C virus, measles virus, rabies virus, and influenza virus. Moreover, using yeast phage display technology has led to producing novel mAbs against HIV-1 [161]. For example, 4Dm2m is a broadly neutralizing CD4-antibody fusion protein that is remarkably successful against HIV-1 [162]. Chen and co-workers enhanced flexibility, stability and half-life of 4Dm2m by introducing numerous modifications into the original molecule by taking advantage of phage-display library technologies and structure-guided design [162].

The **HexaBody** technology by Genmab is based on the observation that IgG antibodies may form ordered hexamers on cell surfaces after binding to their antigen. These hexamers engage the first component of the complement cascade, C1, thereby causing complement-dependent responses. When the antigen binds, conformational changes govern the exposure of the C1 binding site and complement activation [163,164]. de Jong and colleagues discovered a mutation in the Fc region of IgG that favors hexamer formation significantly more quickly upon mAb interaction with the antigen. Consequently, the complement system exerts higher activity levels which result in improved mAb effector functions. This platform has been acknowledged as a safe and effective method for the improvement of mAb activity [165] and is being exploited for optimizing mAb candidates. For instance, Genmab has been developing an antibody against multiple myeloma, named HexaBody-CD38 (GEN3014), which is undergoing Phase 1/2 clinical studies (NCT04824794) in patients affected by hematological malignancies. HexaBody-CD38 demonstrated significant increase in CDC and significant anti-tumor action [9,165]. Another example is represented by the hexabody isoform of mAb 2C7, which binds to a *Neisseria gonorrhoeae* lipooligosaccharide epitope expressed by >95% clinical isolates and hastens gonococcal vaginal clearance in mice [166].

The **DuoBody**^®^ platform by Genmab is another interesting comprehensive technology for the discovery and production of bispecific antibodies that might help with cancer, autoimmune, infectious, and central nervous system disease antibody treatment. As reported above, bispecific antibodies bind to two epitopes on the same or separate targets. This might increase the specificity and effectiveness of the antibodies in inactivating target cells or pathogens [167,168,169]. The FDA has approved the first therapy created using DuoBody^®^ technology platform [170].

Finally, the **DuoHexaBody** platform by Genmab combines the two technologies described above (Hexabody and DuoBody) and was used to develop DuoHexaBody-CD37 (GEN3009), a bispecific antibody that targets two non-overlapping CD37 epitopes. A Phase 1/2 clinical trial in patients suffering from hematologic malignancies is now under way [171]. Overall, HexaBody, DuoBody, and DuoHexaBody represent extremely promising and innovative methodologies which need robust approval and implementation into clinical trials.

IgM antibodies, unlike IgG molecules, already exist as pentameric oligomers held together by covalent bonds. The covalent linkage of IgG monomers via disulfide bonds in an IgM-derived 18 amino acid carboxyterminal extension, and furthermore between cysteine residues inserted at position 309, has been used to activate complement [172,173].

**DNA** and **messenger RNA (mRNA) technologies** have become increasingly popular since the most recent achievements in COVID-19 vaccine design and development. Administration of DNA-encoded mAbs and vaccines is a new and promising avenue that deserves exploration and exploitation, although some concerns must be addressed, such as the need for adjuvants or optimized delivery devices to obtain high immune response efficiency. In the use of DNA vaccines in humans, advances have been achieved using two approaches. The first one includes methods for physical delivery, such as a gene gun [174] or electroporation [175]. These have boosted the immunogenicity of DNA vaccines in human volunteers dramatically [176,177]. The second is the creation of a heterologous prime-boost algorithm [178] whereby the donors are primarily inoculated with a DNA vaccine, then boosted with either recombinant protein antigens or standard dead or live attenuated vaccines [179,180,181]. It is well accepted that DNA immunization can result in high-quality B-cell responses, which can be used to make highly functional mAbs [182,183]. When it comes to developing mAbs against more complex targets, such as membrane proteins, DNA immunization is more effective than traditional protein-based immunization methods [184].

For example, Elliott and colleagues developed synthetic plasmid DNA to encode two new influenza A and B mAbs that are broadly cross-protective [185]. They showed that this method generates strong quantities of functional antibodies directed against influenza A and B viruses in mouse serum via accelerated in vivo delivery of these plasmid DNA-encoded mAb (DMAb) constructs. For the first time, the authors demonstrated that FluA and FluB DMAbs are functionally comparable to recombinant mAbs produced in vitro by standard cell lines. The advantages of the DMAb technology make it a potential delivery method while offering benefits at every step of the supply chain. Indeed, DNA delivery is considerably less expensive than the traditional way of generating mAbs [185]. 

mRNA-based vaccines and therapies represent a new class of revolutionary medications. mRNA-1944, encoding a mAb against Chikungunya virus, has been the first mAb encoded by mRNA to be tested in a human trial [186] and will provide critical information on how mRNA may be employed to produce mAbs systemically in a dose-dependent and accessible way. mRNA-1944 encodes a completely human IgG antibody that was first isolated from B cells of a patient with a history of significant Chikungunya immunity. Within Moderna’s patented lipid nanoparticle (LNP) technology, it comprises two mRNAs that encode the heavy and light chains of this anti-Chikungunya antibody. Clinical investigations using mRNA-1944 demonstrated a linear dose-dependent relationship [186,187]. An injectable formulation containing a lipid nanoparticle–encapsulated mRNA molecule encoding this antibody protected mice against viral infection and triggered protective serum antibody responses in macaques [186]. The in vivo findings of this research opened the way to clinical trials of mRNA-based passive immunotherapy for human Chikungunya infection.

Modern medicine is being transformed by antibody immunotherapy. However, mAbs have certain drawbacks, such as production restrictions, cold-chain storage and transportation requirements, and expensive manufacturing and distribution processes. Transient in vivo gene delivery technologies, such as non-viral synthetic plasmid DNA and messenger RNA vectors built to express tailored mAb genes, might help overcome some of these obstacles. In the case of DMabs or mRNA-encoded mAbs, the body itself operates as a factory for antibody production, thus reducing both costs and the number of procedures necessary in bioprocesses [179].

Both DNA- and mRNA-mAbs have the potential to be used quickly as tools for the control of new infectious diseases. However, one of the advantages of DNA-mAbs over mRNA-mAbs is the speed with which the final formulation for distribution may be achieved. Overall, in silico approaches can improve mAb sequence, which can then be delivered either by a viral vector or as synthetic DNA. While synthetic DNA does not trigger an anti-vector backbone immunological response, viral vector integration within the host genome is possible [180]. On the other hand, mRNA must go through additional processes before being packed and inoculated or delivered. Inside cells, mRNA allows for fast protein expression by skipping the transcription step required when DNA is provided and by directly interacting with the cytoplasmic ribosomes to translate the desired protein. Conversely, DNA-mAbs should first enter the nucleus, be transcribed and then translated into the desired product. In this case, the amount of plasmid that might go into the nucleus is the relevant aspect to be considered. Several investigations have demonstrated that DNA-mAbs have protective effects in mice against various viral pathogens: Dengue virus [188], influenza A and B viruses [185,189,190], Ebolavirus [31,189], Zika virus [190], CHIKV [191], rabies [192], and HIV [193].

By integrating in vitro **somatic hypermutation (SHM)** with mammalian cell display, Peter M. Bowers and colleagues devised a unique approach for the collection and maturation of human antibodies that replicates fundamental characteristics of the adaptive immune system. SHM is dependent on the action of the B cell-specific enzyme activation-induced cytidine deaminase (AID) and can be replicated in non-B cells through expression of recombinant AID [194]. This method addresses many of the earlier constraints of mammalian cell display, allowing for direct antibody selection and maturation as full-length, glycosylated IgGs. Starting with a small number of variable region genes, the immune system has evolved to produce a high frequency of functional antibodies. By directly deaminating cytidine residues in Ig genes, AID is required for the beginning of SHM in B cells [195,196]. To do this, AID is directed towards V-region DNA sequences known as hot spots (e.g., WRCH), which cause mutations and amino acid changes in sites that are biased to alter antigen binding [197]. This method allows for de novo antibody maturation from a naive antibody library, as well as the maturation of preexisting antibodies. SHM affinity maturation in human B cell lymphoma lines has been reported in vitro [198].

**Strategies to enhance antibody effector functions** through Fc engineering represent an extremely promising avenue that deserves attention in the field of antiviral mAb discovery and development [1]. Mutations in the constant part of the mAb can extend antibody half-life: examples are represented by the anti-SARS-CoV-2 VIR-7831 and VIR-7832 mAbs, which contain the M428L/N434S mutations [199], and by the anti-RSV antibody Motavizumab, which is characterized by the M252Y/S254T/T256E substitutions [200]. Additional amino acid replacements, such as L234A, L235A, and P329G, have been found to completely abolish detectable binding to FcRIIa, FcRI, IIB, and IIC for IgG1 and IgG4, thereby preventing antibody-dependent enhancement of disease (ADE) [201]. The GAALIE modification (G236A/A330L/I332E) was shown to favor maturation of dendritic cells and induce protective CD8+ T cell responses by an anti-influenza antibody [202]. Taken together, these reports underline the capacity for IgG antibodies to promote functions which go beyond virus neutralization and encompass the so-called “vaccine-like effects” [1].

**Computational techniques** can predict antibody/antigen structures, engineer antibody function, and build antibody−antigen complexes with superior attributes based on high-throughput sequencing and a growing number of experimental structures of antibodies/antibody−antigen complexes. Numerous in silico approaches can be used to generate effective antibodies. For instance, prediction of (i) antibody−antigen binding, epitope mapping, and affinity maturation, (ii) aggregation, stability, and immunogenicity of antibodies, (iii) antibodies’ allosteric effects, (iv) modulation of the effector functions, and (v) structure prediction of variable domain, complementarity-determining regions, and (vi) vaccine design. Antibody−antigen binding affinities can be improved via in silico modifications of antibody residues. An example is represented by the modification of an anti-lysozyme antibody [203]. Lippow et al. were able to obtain a tenfold increase in affinity by docking the antibody onto its epitope on the antigen’s surface [204]. SnugDock [205] was proposed as a new method for predicting high-resolution antibody-antigen complex structures by physically optimizing the antibody−antigen rigid-body locations concurrently. When the crystal structure of an antibody is not available, this technique is especially advantageous since it allows for flaws in an antibody homology model that would otherwise impede rigid backbone docking predictions. Models of the West Nile virus envelope protein DIII in combination with the neutralizing E16 antibody Fab have been successfully reached by Aroop Sircar and colleagues by applying SnugDock [205]. Sefid and co-workers engineered a VHH nanobody against the Bap antigen in *Acinetobacter baumannii* utilizing in silico modeling [206].

Since the SARS-CoV-2 pandemic began, in silico approaches have played a major role, as represented by antibodies against COVID-19 that, despite the emergence of a number of mutations, can still bind to the virus [207]. For example, docking studies revealed that tixagevimab, bamlanivimab, and sotrovimab can form a stable complex with the Delta variant, while neutralizing the majority of the SARS-CoV-2 Alpha strains. According to the simulations, tixagevimab, regdanvimab, and cilgavimab can successfully neutralize most B.1.1.7 strains, whereas bamlanivimab, tixagevimab, and sotrovimab can effectively suppress Delta. The same study showed that while presently available mAbs might be utilized to treat COVID-19 caused by variants of SARS-CoV-2, chimeric antibodies could provide superior outcomes [207].

Effector functions such as ADCC, CDC, and ADCP can also be engineered using high-throughput computational techniques and the available structures [208]. Structure prediction of variable domain and complementarity-determining regions is one aspect of in silico antibody research that has received a lot of attention, especially from the industry. Chemical Computer Group (CCG), Schrödinger Inc. (New York, NY, USA) [209], and Accelrys Inc. (San Diego, CA, USA) are just a few examples of companies that have created techniques to achieve these goals (e.g., PIGS server [210]). Another way to predict the structure of variable domain and complementarity-determining regions is to use homology modeling [211]. RosettaAntibody uses homology to anticipate and optimize the heavy chain variable domain (VH)/light chain variable domain (VL) [212,213].

## 4. Conclusions

Despite tremendous progress in developing new mAbs, there are still many obstacles to overcome in every step of the developing process, as well as clinical and market challenges such as viral antigenic escape, viral variability, and short duration of viral diseases that makes them commercially less attractive than chronic diseases. Many efforts are required to overcome these challenges, including advancing more potent mAbs, development of new formulation (liquid vs. lyophilized) and delivery (injection vs. aerosol) methods, efficient clinical trials which include mAb combinations, and engagement with organizations operating in low- and middle-income countries in order to favor technology transfer and access to these new bioproducts.

Since the approval of the first murine mAb in 1986, mAb-based therapy has revealed that antiviral mAbs may be used to recruit the endogenous immune systems of infected organisms to induce long-lasting vaccine-like effects and reduce the clinical and economic impact of these infections. The ability to engineer these molecules in order to improve their properties as well as to target intracellular compartments, bind two different antigens simultaneously, deliver drug conjugates, and generate Fc fusions revolutionized the treatment of human diseases, especially viral infections. Out of 104 currently approved mAbs, 6 (5.76%) were approved in the 1990s, 16 (15.38%) from 2000 to 2010, 70 (67.3%) from 2011 to 2020, and 12 (11.5%) in the past two years (2021–2022), thus showing an upward growth in the production and marketing authorization of therapeutic mAbs.

## Figures and Tables

**Figure 1 biomedicines-10-01861-f001:**
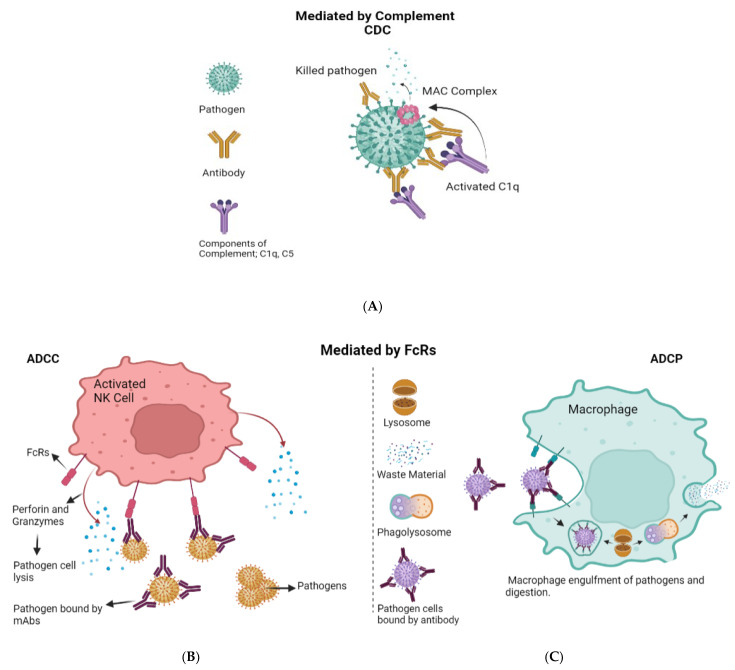
Antibody effector functions. (**A**) Complement-dependent cytotoxicity (CDC). When an antibody binds to an antigen on the cell surface, the complement component C1q is activated and starts the cascade that leads to formation of the C5b-9 membrane attack complex (Mac), which causes cell lysis. (**B**) Antibody-dependent cellular cytotoxicity (ADCC) involves lysis of target cells that have been opsonized by antibodies. In the image shown here, the antibody Fc domain interacts with activated Fc receptors (FcR) on FcR-positive immune cells such as NK cells. This antibody–FcR interaction causes the production of cytokines such as IFN- and cytotoxic molecules such as perforin and granzymes, which induce pathogen cell death. (**C**) In antibody-dependent cell-mediated phagocytosis (ADCP), the interaction of the antibody Fc domain with the activated FcRs on phagocytes causes phagocytes to engulf the opsonized pathogens, resulting in clearance.

**Figure 2 biomedicines-10-01861-f002:**
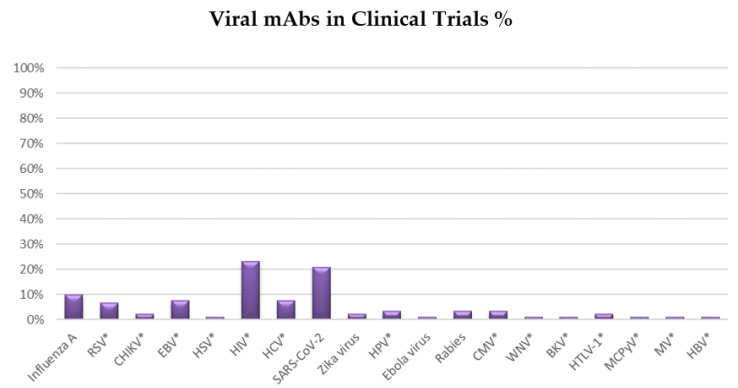
Pie chart summarizing the distribution of clinical trials for mAbs against viral diseases. The graph displays the prevalence of monoclonal antibodies (mAbs) in clinical trials against various viruses. On 23 December 2021, the survey was concluded from https://clinicaltrials.gov/. *Abbreviations and acronyms used: Human immunodeficiency virus (HIV), Coronavirus Disease 2019 (COVID-19), Epstein-Barr Virus (EBV), Hepatitis C Virus (HCV), Hepatitis B virus (HBV), Respiratory Syncytial Virus (RSV), Human Papilloma Virus (HPV), Cytomegalovirus (CMV), Chikungunya virus (CHIKV), Human T-cell lymphotropic virus type 1 (HTLV-1), Herpes Simplex Virus (HSV), West Nile virus (WNV), BK virus (BKV), Merkel cell polyomavirus (MCPyV), Measles Virus (MV).

**Table 1 biomedicines-10-01861-t001:** Anti-SARS-CoV-2 mAbs in clinical studies.

Sponsors	Drug Code	Most Advanced Study	Trial IDs	Est. Start	Est. Primary Completion
Celltrion	CT-P63	Phase 1 pending	NCT05017168	Sep 2021	Oct 2021
Exevir Bio BV	XVR011	Phase 1	NCT04884295	Aug 2021	Sep 2021
Jemincare Group	JMB2002	Phase 1	ChiCTR2100042150	NA	NA
Luye Pharma Group Ltd.	LY-CovMab	Phase 1	NA	NA	NA
AbbVie	ABBV-47D11	Phase 1	NCT04644120	27 Nov 2020	Aug 2021
HiFiBiO Therapeutics	HFB30132A	Phase 1	NCT04590430	Oct 2020	Jul 2021
Ology Bioservices	ADM03820	Phase 1	NCT04592549	4 Dec 2020	Sep 2021
Beigene	DXP604	Phase 1	NCT04669262	15 Dec 2020	May 2021
Zydus Cadila	ZRC-3308	Phase 1 pending	NA	NA	NA
Hengenix Biotech Inc	HLX70	Phase 1 pending	NCT04561076	9 Dec 2020	Sep 2021
CORAT Therapeutics	COR-101	Phase 1/2	NCT04674566	31 Jan 2021	Oct 2021
Vir Biotechnol./	VIR-7832	Phase 1/2	NCT04746183	31 Jan 2021	Nov 2021
AbCellera/Eli Lilly and Company	LY-CoV1404, LY3853113	Phase 2	NCT04634409	NA	Aug 2021
Sorrento Therapeutics, Inc.	COVI-AMG (STI-2020)	Phase 2	NCT04734860	April 2021	Sep 2021
Beigene	DXP593	Phase 2	NCT04532294; NCT04551898	31 Aug 2020; 30 Oct 2020	15 Oct 2020; 28 Feb 2021
Junshi Biosciences/Eli Lilly and Company	JS016, LY3832479, LY-CoV016	Phase 2	NCT04441918; NCT04441931; NCT04427501	5 Jun 2020; 19 Jun 2020; 17 Jun 2020	Dec 2020; 2 Oct 2020; 11 Mar 2021
Mabwell (Shanghai) Bioscience Co., Ltd.	MW33	Pivotal Phase 2	NCT04533048; NCT04627584	7 Aug 2020; Nov 2020	Dec 2020; May 2021
Toscana Life Sciences Sviluppo s.r.l.	MAD0004J08	Phase 2/3	NCT04932850; NCT04952805	March 2021; June 2021	Oct 2021; March 2022
Bristol-Myers Squibb, Rockefeller University	C144-LS and C-135-LS	Phase 2/3	NCT04700163; Activ-2 study	11 Jan 2021; TBD	June 2021; TBD
Sinocelltech Ltd.	SCTA01	Phase 2/3	NCT04483375; NCT04644185	24 Jul 2020; Mar 2021	Nov 2020; Nov 2021
Adagio Therapeutics	ADG20	Phase 2/3	NCT04805671 NCT04859517	Mar 2021; Apr 2021	Dec 2021 July 2022
Brii Biosciences	BRII-196	Phase 3	NCT04479631; Activ-3 study	12 Jul 2020; TBD	Mar 2021; TBD
Brii Biosciences	BRII-198	Phase 3	NCT04479644; Activ-3 study	13 Jul 2020; TBD	Mar 2021; TBD
Tychan Pte. Ltd.	TY027	Phase 3	NCT04429529; NCT04649515	9 Jun 2020; 4 Dec 2020	Oct 2020; 31 Aug 2020
AstraZeneca	AZD7442 (AZD8895 + AZD1061)	Phase 3	NCT04507256; NCT04625725; NCT04625972	17 Aug 2020; 17 Nov 2020; 16 Nov 2020	Sep 2021; Feb 2022; Jan 2022
Celltrion	CT-P59	EUA #	NCT04525079; NCT04593641; NCT04602000	18 Jul 2020; 4 Sept 2020; 25 Sept 2020	Nov 2020; 23 Dec 2020; Dec 2020
Vir Biotechnol./ GlaxoSmithKline	VIR-7831/ GSK4182136	EUA *	NCT04545060; Activ-3 study	27 Aug 2020; TBD	Jan 2021; TBD
AbCellera/Eli Lilly and Company	LY-CoV555 (LY3819253); combination of LY-CoV555 with LY-CoV016 (LY3832479)	EUA * for bamlanivimab/etesevimab combination therapy	NCT04411628 (Phase 1); NCT04427501 (Phase 2); NCT04497987(Phase 3); NCT04501978 (Activ-3 study); NCT04518410 (Phase 2/3)	28 May 2020; 13 Jun 2020; 2 Aug 2020; 4 Aug 2020; Aug 2020	23 Aug 2020; 15 Sept 2020; 8 Mar 2021; July 2021; Feb 2021
Regeneron	REGN-COV2 (REGN10933 + REGN10987)	EUA *	NCT04425629 (Phase 1/2); NCT04426695 (Phase 1/2); NCT04452318 (Phase 3)	16 Jun 2020; 10 Jun 2020; 13 Jul 2020	19 Dec 2020; 25 Jan 2021; 15 Jun 2021

* EUA: emergency use authorization granted in the US. # EUA: emergency use authorization granted in South Korea. NA: not available.

## Data Availability

Not applicable.

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
