# Peer review of "Recent Progress in the Discovery and Development of Monoclonal Antibodies against Viral Infections"

_biomedicines, 2022, doi:10.3390/biomedicines10081861_

Round 1
Reviewer 1 Report
Overall this is an excellent and extremely well-written and well-referenced review of the current status and approach utilized in development of mAbs against viral pathogens.
Reviewer 2 Report
Pardis Mokhtary et al. reviewed important findings regarding recent progress in drug discovery and development of monoclonal antibody against viral infections.
Points to be considered
1) The rationale of why the authors came up with this review.
2) What is the information that is not exactly available that motivated the authors to come up with this information. What are the current caveats and how do the authors highlight the current research in answering them? If not they need to address in future directions.
3) How would the authors address the "garbage in garbage out" potential bias while performing a bioinformatic analysis?
4) How Would the authors address the «frame problem»(the difficulty in identifying and updating a set of axioms to properly describe the environment for autonomous agents?
5) This reviewer personally misses some important information regarding aspects related to infections but partially neglected so far: as an example, the pandemic of coronavirus disease 2019 (COVID-19) caused by the novel severe acute respiratory syndrome coronavirus 2 (SARS-CoV-2) represents an extraordinary challenge to world’s scientists. High infectivity of the virus, lack of effective antivirals and vaccines, and large asymptomatic populations make the disease hard to conquer. The clinical spectrum of COVID-19 appears to be abnormally wide, and its pathogenesis is yet to be decoded. Patients can experience a range of clinical manifestations, from no symptoms to critical illness. Accurate prediction of clinical outcomes for patients across this spectrum is often difficult. Advanced age and comorbidities are associated with more severe disease and poorer outcomes. In severe cases, it seems that most organ damage is done through an immune-mediated mechanism, although SARS-CoV-2 is the necessary initiator. Treatment for COVID‐19 is currently only supportive, focused on appropriate management of the associated respiratory dysfunction. The synergistic role that viral and host-dependent mechanisms play in the disease pathogenesis suggests that new therapeutic strategies must combine antiviral drugs and immune-modulating agents.
7) In the frame of point 6 thinking, (1) Are patients with infections at higher risk of developing acute or late-onset endocrine diseases or dysfunction? (2) May the underlying endocrine diseases or dysfunctions be considered risk factors for poor prognosis once the infection has occurred? (3) Are there defined strategies to manage endocrine diseases despite infection agents-related constraints and thus impact on the authors findings? (please refer to PMID: 34209964 and expand).
6) The authors need to highlight what new information the review is providing to enhance the research in progress.
Reviewer 3 Report
This review about antibody therapeutics for infectious viral diseases focuses on several viruses like Zika virus, Influenza virus, MERS, etc, then reviewed several antibody discovery technologies. In my perspective, several more aspects of mAb therapeutics should be included.
- Strategies to enhance antibody effector function should be reviewed. For instance, SARS-CoV-2 VIR-7831 and VIR-7832 antibodies were mutated to contain an “LS” mutation in the Fc region to prolong antibody half-life and potentially enhance distribution to the respiratory mucosa. Also, The VIR-7832 antibody additionally contains an Fc GAALIE mutation shown to boost T-cell immunity. GAALIE is an acronym for the amino acid mutations in the Fc region, specifically G236A, A330L, and I332E. These mutations would enable antibody-dependent complement-mediated cytotoxicity without inducing antibody-dependent enhancement. In other words, it would neutralize the virus and clear out infected cells faster. There might be other strategies to enhance effector function. Here’s a reference:
Pelegrin M, Naranjo-Gomez M, Piechaczyk M. 2015. Antiviral Monoclonal Antibodies: Can They Be More Than Simple Neutralizing Agents? Trends in Microbiology 23:653-665.
- Antibody discovery also includes B cell immortalization method and this should be reviewed as well.
- Alpaca nanobody exhibits extreme superiority than mAbs and should be briefly discussed as well.
- Figure 2: Please try to list each virus inside the barchart as the legend color is confusing
Reviewer 4 Report
This article reviews some of the recent developments in mAb discovery against viral infections and illustrates how mAbs can help to combat viral diseases and out breaks, which would provide better understanding of mAb therapy for viral diseases. The work is well organized and comprehensively described,and worthy of publication in Biomedicines after minor revision.
---Fig. 2: “Coronavirus Disease 2019 (COVID-19)” was used in Fig.2; in line 86 the authors described “including HIV, SARS-CoV-2, Ebola, as reported in Figure 2”. However, in figure legends of Fig.2, the author described “SARS-CoV-2 is not included in the design of this pie chart”, so the description is a little confused. Moreover, the content in Figure 2 should be briefly explained in the text.
---In “3. Standard and new technologies for the discovery and development of highly effective mAbs”, more details focusing on the method itself might be provided, such as the advantages and disadvantages of the hybridoma method, the phage display and so on.
